# Problematic Gaming during COVID-19 Pandemic: A Systematic Review, Meta-Analysis, and Meta-Regression

**DOI:** 10.3390/healthcare11243176

**Published:** 2023-12-15

**Authors:** Chiara Imperato, Alessandro Giardina, Tommaso Manari, Antonio Albano, Christian Franceschini, Adriano Schimmenti, Alessandro Musetti

**Affiliations:** 1Department of Humanities, Social Sciences and Cultural Industries, University of Parma, 43121 Parma, Italy; chiara.imperato@unipr.it (C.I.); tommaso.manari@unipr.it (T.M.); 2Institute of Psychology, University of Lausanne, 1015 Lausanne, Switzerland; alessandro.giardina@unil.ch; 3Department of Medicine and Surgery, Faculty of Medicine and Surgery, University of Parma, 43126 Parma, Italy; antonio.albano@studenti.unipr.it (A.A.); christian.franceschini@unipr.it (C.F.); 4Faculty of Human and Social Sciences, UKE—Kore University of Enna, 94100 Enna, Italy; adriano.schimmenti@unikore.it

**Keywords:** problematic gaming, COVID-19, meta-analysis, prevalence

## Abstract

The COVID-19 pandemic led to government measures enforcing isolation in order to mitigate the spread of the virus. Consequently, online activities, including gaming, increased during this challenging period. Thus, it was possible that problematic gaming (PG) patterns also increased. In this systematic review and meta-analysis, we estimated the prevalence of PG during the COVID-19 pandemic and examined differences among subpopulations. The evaluation of 38 studies revealed that the overall prevalence of PG during the COVID-19 pandemic was 3.6%. Furthermore, higher PG scores were found in undergraduate and gamer subpopulations, as well as in studies using the Gaming Addiction Scale. Finally, meta-regression analyses suggest that stricter government measures, as identified by the Government Stringency Index, may have contributed to a lower prevalence of PG behaviors. A potential explanation of this finding is that containment measures had a protective function with respect to emotional distress, and thus towards PG; alternatively, it could be that current measures for PG become less precise if an individual’s functioning is already impaired due to other reasons, such as COVID-19 restrictions. Further theoretical, methodological, and practical implications of the findings are discussed.

## 1. Introduction

The COVID-19 virus (i.e., SARS-CoV-2) emerged in Wuhan, China, at the beginning of December 2019, and resulted in the declaration of a global pandemic by the World Health Organization (WHO) on 11 March 2020 [1]. During the COVID-19 outbreak, several countries implemented restrictive measures, encouraging people to stay at home in order to contain the spread of the virus. These quarantine measures combined with concerns about the risk of exposure to the COVID-19 virus led to significant consequences on individuals’ mental health [2]. Indeed, accumulated evidence has shown that the COVID-19 pandemic has generated or worsened psychological symptoms [3], including depression and anxiety [4], and stress [5]. In such a challenging period, online activities offered some relief, helping both adolescents and adults to stay in contact with loved ones and mitigate the emotional impact of isolation measures in several meaningful ways [6,7]. For example, Barr and Copeland-Stewart [8] documented a change in how and why people played games during the pandemic and the ways it affected well-being. Specifically, they found that gamers shifted from single-player to multiplayer and from offline to online gaming, preferring more relaxing and comforting games among both new and favorite old ones. As for the reasons and benefits obtained by such changes in habits during the pandemic, the authors pointed out various mental health improvements, such as socialization, cognitive stimulation, increased agency, and a sense of normality in a very extraordinary period. In addition, video games seem to have provided a means for escapism. In this direction, Boldi et al. [6] highlighted via thematic analyses four ways video games contributed to the reshaping of daily life in 330 gamers. In their study, gaming served to: (1) Face the repetitive and void time of the lockdown by restoring a structured schedule, or by being thrown back to infant or adolescent times (temporal escapism); (2) Alleviate strong negative emotions through a feeling of safety experienced in the virtual environment, or revive an emotionally flat day with new challenges of games (emotional escapism); (3) Maintain the contact with loved ones and share sociability without the risk of infection for oneself or others, and overcoming the mandatory co-habitation (social escapism); and (4) Overcome the lack of movement by traveling the virtual spaces of old and new games (spatial escapism). Put differently, video games replaced lost daily routines and helped people deal with unsatisfying aspects of their lives during the lockdown. Other ways gaming helped individuals cope with emotional distress during the pandemic include providing a means to meet social and emotional needs in compliance with the existing measures and free from fears related to COVID-19 [9,10]. Finally, it seems that video games have helped adolescents to increase their sense of self-efficacy and to assume a positive attitude and problem-solving coping style towards the lockdown [11].

### 1.1. Problematic Gaming during COVID-19

The benefits of gaming during the pandemic seem to be particularly valid for individuals committed to gaming yet in a healthy and social way [7,12]. However, problematic gaming behaviors can also occur. For instance, previous studies [13] underlined that the temporary escape provided by video games can lead to excessive use and ultimately disordered use, and this was even more true for individuals who have difficulty regulating their emotions, for those high on maladaptive personality traits and psychopathological symptoms. Therefore, problematic gaming (PG) has gained increased attention during the global quarantines connected to COVID-19.

#### 1.1.1. Problematic Gaming Definitions and Criteria

From a symptom-based perspective, PG was defined in 2013 as Internet Gaming Disorder by the Diagnostic and Statistical Manual of Mental Disorders (DSM-5 [14]) and is considered a potentially eligible condition to be included in forthcoming editions of the manual. This approach transposed the criteria adopted to diagnose substance use disorders into gaming, thus defining PG by the development of (1) a preoccupation with gaming; (2) an increasing amount of time spent playing to satisfy the urge; (3) withdrawal symptoms; (4) continued gaming despite the problems caused; (5) multiple attempts to reduce the time spent gaming failed; (6) the use of gaming to improve moods and (7) impairments in life functioning because of gaming (jeopardizing school or work, conflict with family members, etc.). In the recently published DSM-5-TR [15], the status of Internet Gaming Disorder has been unchanged. In 2019, the WHO also included Gaming Disorder in the International Classification of Disease (ICD-11 [16]), under the category of Disorders due to substance use or addictive behaviors section. Adopting a more streamlined approach, the WHO defined Gaming Disorder as a persistent pattern of uncontrolled gaming that impairs functioning in various domains.

#### 1.1.2. Changes in Problematic Gaming during COVID-19

A question was raised about whether PG increased during the COVID-19 outbreak [17]. For instance, Teng et al. [18] longitudinally analyzed both video-game use and PG during the COVID-19 pandemic in a sample of children and adolescents. The authors found that depression and anxiety before COVID-19 generated a significant increase in both video-game use and PG during the pandemic. Therefore, the authors argued that PG represented a dysfunctional compensation resulting from poor psychological health due to the pandemic situation, in line with the general compensatory hypothesis of problematic Internet use [19]. Furthermore, by comparing levels of emotional distress in two independent groups of gamers before and during the establishment of self-confinement measures related to COVID-19, Giardina et al. [7] found that emotional distress predicted more strongly PG during the pandemic, as the level of gaming-related relaxation increased. Taken together, these findings suggest that individuals already at risk because of their psychological vulnerability may have been involved with gaming to find temporary relief from the pandemic situation, yet resulting in compulsory gaming patterns in the long run. In this vein, it is crucial to understand to what extent the protracted isolation due to COVID-19 impacted individuals’ PG, bringing the most vulnerable ones to game excessively and problematically.

### 1.2. Objectives of the Present Systematic Review and Meta-Analysis

The present systematic review and meta-analysis aims to explore the prevalence of PG during COVID-19, also analyzing differences among subpopulations. To date, four systematic reviews examined the role of PG during the COVID-19 pandemic. A systematic review by Masaeli and Farhadi [20] showed that Internet-based addictive behaviors during the first months of the COVID-19 pandemic (i.e., articles published until October 2020) actually increased, in association with financial hardships, isolation, problematic substance use, and mental health issues such as depression, anxiety, and stress. A systematic review by Pallavicini et al. [21] examined studies published in the early stages of the COVID-19 pandemic, showing that gaming was negatively associated with stress, anxiety, depression, and loneliness, thus mitigating the negative consequences of restrictions due to the COVID-19 pandemic. However, when it comes to gaming disorder, authors pointed out that it has risen, especially among males in their youth. In their systematic review and meta-analysis, Alimoradi et al. [22] estimated the prevalence of different potentially addictive behaviors during the COVID-19 pandemic. As far as gaming addiction is concerned, these authors found that it increased during COVID-19 compared to periods before COVID-19. However, the authors also pointed out that the differences in methods adopted to collect data likely influenced such results. Lastly, in their systematic review, Oceja et al. [23] found a small number of studies on PG during COVID-19, which led to conflicting and inconclusive results due to the heterogeneity of methods.

#### Significance of the Present Systematic Review and Meta-Analysis

Xu et al. [24] suggested that COVID-19 encouraged a rapid spread of information, which may have affected the quality of studies on these topics. Therefore, conducting a systematic review and meta-analysis two years after the pandemic may provide more accurate results from a larger number of studies. On these premises, the aim of this study is to provide an update about PG during the COVID-19 pandemic, estimating the prevalence of PG during COVID-19. In addition, we are interested in analyzing differences in the prevalence of PG among different sub-groups (e.g., study location, type of population, instrument used), also testing which variables (e.g., age, female/male ratio, days since the beginning of COVID-19 pandemic, Global Stringency Index, new cases per million people, and new deaths per million people) predicted the proportion of PG found. Differently from existing systematic reviews and meta-analyses, this study will thus: (1) specifically focus on the role of PG without other forms of Internet-based addictive behaviors or different outcome variables (i.e., stress, anxiety, depression, etc.); (2) include a large number of studies (i.e., longitudinal, experimental, or cross-sectional studies) examining PG during COVID-19; (3) include studies regardless of the measure used to evaluate PG; (4) include studies conducted during all the stages of COVID-19 pandemic.

## 2. Methods

The present systematic review and meta-analysis were conducted following the Preferred Reporting Items for Systematic Reviews and Meta-Analysis (PRISMA) updated statement and guidelines [25]. Furthermore, to increase transparency in academic research, this study was preregistered on the International Prospective Register of Systematic Reviews (PROSPERO) international database (Protocol ID: CRD42022339963) in June 2022. The updated PRISMA checklist is available as Appendix A.

### 2.1. Information Sources and Search Strategy

A systematic search of the existing literature was performed on 23 June 2022, in the following online databases: PubMed, Science Direct, Web of Science, Scopus, MEDLINE (accessed through the EBSCO host platform), and Google Scholar. The search terms were discussed amongst the study team and included a combination of the relevant elements identified for the research questions. The default search string, adapted for each database, was: (COVID-19 OR coronavirus OR 2019-ncov OR SARS-CoV-2 OR lockdown) AND (gaming OR videogame*), restricted within Titles, Abstracts, and Keywords. The database search filter was set to published articles from December 2019 to June 2022 and no further filter was applied (see Appendix A for the specific search strings used in each database). Furthermore, the reference lists of already published systematic reviews, meta-analyses, and relevant articles were examined to detect additional potential results (backward search process). The literature search was updated on 5 December 2022, following the same criteria. For those studies that did not report relevant data for the meta-analysis, the corresponding authors were contacted in order to obtain the missing data (Appendix A).

### 2.2. Eligibility Criteria

Studies were eligible for inclusion in both systematic review and meta-analysis if all the following inclusion criteria (IC) were met: (IC1) focused on online and offline video games, with no restriction regarding the platform or the game genre played; (IC2) use of standardized PG and/or gaming disorder measures; (IC3) conducted during the COVID-19 pandemic (i.e., from December 2019 to the day of the systematic database research). All study designs were considered except for case reports. Studies were excluded if they met at least one of the following exclusion criteria (EC): (EC1) case reports, systematic reviews or meta-analyses, theoretical papers, commentaries, editorials, or published conference proceedings; (EC2) papers not written in English; (EC3) did not specifically focus on video gaming and problematic video gaming (e.g., on the use of the Internet or social media).

Furthermore, we included in the systematic review and not in the meta-analysis the studies not reporting sufficient data to compute statistical analysis.

### 2.3. Selection Process

In order to enable an easier double-blinded review and selection of the references, the retrieved database searches were exported to the web application Rayyan (https://www.rayyan.ai/, accessed on 1 July 2022). After duplicate removal, one author (T.M.) performed preliminary scrutiny on titles and abstracts to exclude irrelevant references. Subsequently, two different authors (A.M. and A.G.) independently assessed the full texts of the remaining studies for inclusion in the systematic review. Results were compared and disagreements were resolved by mutual consensus. Further information on the selection process and results is provided in Figure 1.

To support open science and reproducibility of the results, the full lists of excluded references, as well as all database extracted results, files, and scripts used for the meta-analysis, are publicly available on the Open Science Framework (OSF) online archive: https://osf.io/uvwy9.

### 2.4. Risk of Bias Assessment

The Newcastle–Ottawa Scale [26,27] for assessing the quality of studies in the meta-analysis was adopted. This instrument comprises seven items and three main sections: (i) selection, which includes questions regarding the representativeness of the cases or the adequacy of the sample, (ii) comparability, i.e., the comparability of different outcome groups, and (iii) outcome, which indicates the quality and adequacy of the outcome and its assessment. A total score, ranging from 0 to 10, indicates the overall estimated study quality, with higher scores corresponding to a higher study quality. Two authors (T.M. and A.A.) assessed the risk of bias and disagreements were resolved through discussion. The obtained results are summarized in Table 1 and in detail as Appendix A.

### 2.5. Data Collection and Statistical Analyses

A synthesis of the data was outlined through a description for each included study on the following variables: authors, study location (e.g., Europe, Asia), type of population (i.e., gamers, general population, students—who attended elementary, middle, and high schools, undergraduates—who are enrolled in university but not yet graduated), sample size, PG related variable, female percentage, age range, period in which the study was conducted, and PG scale. Furthermore, the following COVID-19-related measures were taken into consideration: Government Stringency Index (GSI; i.e., a composite measure based on nine indicators of public places closures, bans, and restrictions, with a higher score indicating stricter responses on a scale from 0 to 100 [87]), number of COVID-19 cases per million, and new COVID-19 related deaths per million (retrieved from the website: http://ourworldindata.org, accessed on 1 July 2022).

Statistical analyses were performed using R, version 4.2.2, using “meta” [88] and “dmetar” [89] packages. We estimated the pooled raw proportion of the subjects with PG during the COVID-19 pandemic. The details of the meta-analytical method were: random intercept logistic regression model [90], a maximum-likelihood estimator for tau2, Hartung–Knapp adjustment for random effects model [91,92], logit transformation [90], Clopper–Pearson confidence interval for individual studies [93], and continuity correction of 0.5 in studies with zero cell frequencies.

Publication bias was assessed by a rank correlation test of funnel plot asymmetry [94] and a linear regression test of funnel plot asymmetry [95]. Since we did not obtain statistically significant results, it was not necessary to apply the trim-and-fill method [96] to adjust for funnel plot asymmetry.

We quantified between-study heterogeneity by tau2 and I2 statistics [97], where a value of more than 75% reflects high heterogeneity. We tested the heterogeneity by applying the Wald-type test and Likelihood-Ratio test to Cochran’s Q [98].

In order to explain the heterogeneity, we undertook four subgroup analyses [99] based on the following categorical variables: study location (i.e., America, Asia, Europe, Middle East, World), type of population (i.e., gamers, general population, students, undergraduate), specific instrument used to estimate gaming disorder and risk of bias (i.e., high, low).

Afterward, we tested the moderators in six meta-regressions under random-effects models to check if the raw proportion could be predicted by the following continuous variables: average age, female percentage, days (i.e., number of days between the central day of the research and 31 December 2019; this variable was included in order to understand how many days elapsed between the study and the COVID-19 outbreak, and 31 December 2019, was chosen as the date when the first cases of COVID-19 were recorded), Global Stringency Index (GSI), new cases per million people, and new deaths per million people. Finally, we tested 64 fitted models in a multimodel inference [99] to understand the relative importance of the continuous predictors.

## 3. Results

### 3.1. Overview of the Studies

A total of 2875 records were detected through database searching. After the removal of 1004 duplicates, one author (T.M.) inspected titles and abstracts of 1871 records, of which 147 were selected for full-text scrutiny by two authors (A.M. and A.G.). The final step included 61 records assessed for eligibility. Specifically, a total of 61 studies were included in the systematic review, whereas a total of 38 studies were included in the meta-analysis. The selection list, reasons for exclusions, and the screening steps are synthetically provided in Figure 1. The complete record of the excluded studies and the reason for exclusions is listed as Appendix A (see Appendix A for the complete list of included records, and Appendix A for the complete list of the excluded ones). On 5 December 2022, a second systematic search was conducted to collect any studies published since the first systematic search. Two studies [32,38], added to the previous 61, were then considered eligible.

### 3.2. Study Characteristics

The final sample and its detailed characteristics are presented in Table 1. All the included studies adopted a cross-sectional design and involved 27 countries worldwide. Specifically, 31 were conducted in Asia (China, Hong Kong, India, Indonesia, Japan, Malaysia, Nepal, South Korea, Taiwan, Vietnam), 17 were in Europe or neighboring countries (Finland, Germany, Hungary, Italy, Spain, Sweden, Switzerland, United Kingdom), eight were in the Middle East or neighboring countries (Iraq, Israel, Pakistan, Saudi Arabia, Turkey, United Arabs Emirates), three studies were conducted in North or South America (Brazil, USA), one was from Oceania (New Zealand), while three were cross-national studies. Sample sizes ranged from 77 to 51,246, with a proportion of female participants of 49.19% (information not available for three studies). The mean age of the participants across the studies was 20.66 years old, ranging from 6 years old to 90 years old. Five studies included only gamers, four studies focused on undergraduates, 22 were carried on a general population, and 32 studies comprised students.

Table 1 also includes a full list of the employed instruments. The most used instrument was the Internet Gaming Disorder Scale based on the DSM 5 criteria, in its different versions. Specifically, the Internet Gaming Disorder Short Form (IGDS-9-SF [100]) was employed in 17 studies, 9 studies used the 9-item Internet Gaming Disorder Test (IGDT-9 [101]) and 4 studies adopted the Internet Gaming Disorder Test, the 10-item version (IGDT-10 [102]). The Game Addiction Scale (GAS-7 [103]) with its adapted or extended versions, was used in a total of 8 studies. Other instruments included the Maladaptive Game Use Scale [55], the Video Game Addiction Scale for Children [104], the Mobile Social Online Gaming Addiction Scale [105], the Screener for Substance and Behavioral Addictions (SSBA [106]), the Video Gaming Scale (VGS [107]), the Gaming Disorder Scale for Adolescents—(GADIS-A [108]), and others created instruments ad-hoc.

### 3.3. Risk of Bias Assessment

In order to assess the risk of bias and evaluate the quality of the articles included in the meta-analysis, the Newcastle–Ottawa Scale for cross-sectional studies was adopted. The related findings are reported in Appendix A. The total score (the maximum obtainable value was 10) for each study ranged from a minimum of 3 to 7. Two authors (T.M. and A.A.) evaluated independently all records included, and the inter-judge agreement was 91.32%. Overall, the samples were deemed representative (question 1), however, justification of the sample size was not always provided (question 2). Most of the studies (*n* = 22) did not sufficiently describe non-respondents or the non-response rate (question 3). Outcome variables were measured with validated instrument tools (question 4). Furthermore, the outcomes were assessed with self-report instruments (question 5). The statistical tests used to analyze the data (question 6) were found to be appropriate and adequately justified in 22 studies.

### 3.4. Meta-Analysis

The meta-analysis was performed on 38 studies, from which 41 records were extracted (*K* = 41, *n* = 92,705). The proportion of gaming disorder was 3.6% (95% CI, 0.03–0.05), with significant and high heterogeneity (*Q* = 3210.88, *df* = 40, I^2^ = 98.8%, *p* < 0.001) (see Figure 2 for funnel plot and Appendix A for the events). The Begg and Mazumdar’s (*p* = 0.062) and Egger’s (*p* = 0.574) tests for publication bias were both statistically non-significant.

#### 3.4.1. Subgroup Analyses

Further analyses were carried out in the following subgroups: study location, categories of participants, rating scale, and risk of bias (Table 2). The subgroups of study location and risk of bias showed statistically non-significant results, whereas the categories of participants (*p* < 0.001) and the rating scale used (*p* < 0.001) were statistically significant. Regarding the categories of participants, undergraduates represented a proportion of 0.10 (95% CI, 0.05–0.21), followed by gamers (0.08; 95% CI, 0.03–0.22), students (0.03; 95% CI, 0.02–0.07) and lastly general population (0.02; 95% CI, 0.01–0.04). The rating scales used to assess gaming habits delineated a proportion of 0.08 for the GAS (95% CI, 0.06–0.13), 0.04 for other instruments (95% CI, 0.01–0.24), 0.04 for the tools based on the DSM 5 criteria (95% CI, 0.02–0.08), and 0.02 for the IGDS (95% CI, 0.01–0.04).

#### 3.4.2. Meta-Regression Analyses

Meta-regression analyses were further performed in order to verify whether the meta-analytic results were predicted by the following variables: average age, gender distribution, days passed since the beginning of the COVID-19 outbreak (i.e., calculated as the central day in the range of days provided), Global Stringency Index (GSI), new cases per million people, and new deaths per million people (Table 2). All meta-regression analyses were statistically non-significant, except for the GSI (*F* = 8.62, *p* < 0.01). Gaming disorder was significantly predicted by GSI, with a negative slope (Coef. = −4.48).

## 4. Discussion

The objective of the current study was to provide an up-to-date and comprehensive systematic review and meta-analyses regarding the estimated prevalence of PG during the COVID-19 pandemic, including an analysis of the subpopulations most impacted and the effect of government restrictions. To this aim, we analyzed 62 records, of which 38 (for a total of 41 studies) were also included in the meta-analysis, comprising a total of 92,705 participants.

Results showed that the overall prevalence of PG during COVID-19 was 3.6%, which is a bit higher than the findings of a meta-analysis conducted before the COVID-19 pandemic (i.e., a prevalence of 3.1% [109]). When it comes to sub-group analysis, we found that the proportion of PG significantly differed based on the type of population considered and the measures used in the studies. Specifically, the proportion of PG was higher among undergraduates and gamers, while among students and the general population, the proportion of PG during COVID-19 was lower. In a recent report, the American Psychological Association [110] identified young adults aged between 18 and 23 as a vulnerable group characterized by high levels of stress and depressive symptoms, especially when compared to older populations. Moreover, this is a population for which the freedom of movement between the family and the academic/professional contexts is fundamental for the creation of a role in society, and thus for healthy development. In this sense, the COVID-19 restrictions may have prevented students living abroad from returning home, or students living with their parents from having a satisfying academic life in front of an uncertain future. In line with Hidalgo-Fuentes et al.’s [111] review, social factors, such as social support, are particularly important in addressing problematic technology use. In this distressing scenario, higher levels of PG may be explained by the need to regain some control over spatial movements—i.e., to reduce the distance from significant others—or to relieve emotional distress [6,112]. Eventually, a higher prevalence of PG was found among gamers. Not surprisingly, for this population, gaming was an “at hand” solution to deal with emotional distress; thus, for the most vulnerable players, the COVID-19 restrictions could have fostered an excessive involvement with games to seek short-term relief [7].

As for the measure used, studies using the GAS showed the highest prevalence of PG. As King et al. [113] pointed out, there are many different measures used to screen and assess PG, and such heterogeneity of tools hinders comparisons across studies. Indeed, the variety of measures used was more than just a methodological concern, given that the theoretical frameworks and the coverage of DSM5 and ICD-11 criteria were different across tools. Differences among criteria considered could bring to over- or underestimation of PG, thus leading to different conclusions about its prevalence. Nonetheless, a meta-analysis in the field [114] analyzing five measures used to assess PG (i.e., Gaming Addiction Scale, Adolescent Internet Addiction Scale, Internet Gaming Disorder Test-10, Lemmens Internet Gaming Disorder Scale-9, Internet Gaming Disorder Scale-9 Short Form) found that the PG scales showed good reliability overall, and therefore the reliability of the PG assessment tools can be generalized.

As far as meta-regression results are concerned, we found that the prevalence of PG changed as a function of the GSI, that is, the extent to which the countries adopted stringent responses following the COVID-19 outbreak. The relationship between PG and GSI was negative, such that the fewer the measures adopted by the countries, the higher was PG prevalence. While these results may seem counterintuitive, a previous meta-analysis by Scarpelli et al. [115] also showed better sleep quality in countries with a greater GSI during the COVID-19 pandemic. The authors suggested that the stringency of governmental measures could have made people feel safer during the pandemic, thus reducing anxiety and uncertainty. On the other hand, to face daily uncertainty due to the COVID-19 pandemic in countries in which measures were not so stringent could have made people vulnerable and therefore more easily exposed to PG.

An alternative explanation for this result lies in the way PG is conceptualized and measured. Indeed, the core aspect of this construct is the impairment of different domains of life due to the continued use of gaming. However, if those domains are already impacted by another circumstance (i.e., restrictions due to COVID-19), available instruments for PG may not be capable of detecting the degree and quality of the attachment to the game of an individual. For example, the question “Have you jeopardized your school activities because of gaming?” would likely receive a negative answer during the COVID-19 restrictions, essentially because there were no school activities to jeopardize.

Taken together, our results have twofold implications. From a methodological point of view, they support the need for an agreement regarding PG symptoms measurement [116] and the need for more explorative and process-based approaches to evaluate the quality of the attachment to gaming [117]. From a practical point of view, results seem to suggest that public health efforts should be targeted toward undergraduate populations, specifically in countries where the measures adopted during COVID-19 were less stringent.

The present systematic review and meta-analysis come with some limitations. Firstly, the sub-group analysis and meta-regressions we considered did not explain the high percentage of variance the overall results showed, indicating that the prevalence of PG could vary between different sub-groups not considered in the present systematic review and meta-analysis. Furthermore, the heterogeneity of tools measuring PG did not allow a more accurate meta-analytic assessment of possible antecedents or outcomes of PG during COVID-19. In addition, studies used self-reported measures and therefore could be affected by biased recall or denial, or defensiveness mechanisms, also considering the extraordinary nature of the pandemic period which may have led people to misperceptions. Also, it must be noted that some of the sub-group analyses performed involved groups with small frequencies (e.g., study location, World k = 2), and such small frequencies could impact the statistical power. Lastly, despite the majority of scientific literature being published in English, relevant studies published in different languages may have been excluded.

## 5. Conclusions

In the present systematic review and meta-analysis of 38 studies on the prevalence of PG during the COVID-19 pandemic, we highlighted that the type of populations most affected by PG were undergraduates and gamers. Moreover, the type of instrument used to assess PG and the government measures also affected the prevalence of PG. Differently from other reviews conducted on this topic, the present systematic review and meta-analysis focus exclusively on PG, additionally providing an update on the prevalence of PG during the COVID-19 pandemic two years after its spread, and therefore including studies conducted during all the stages of the COVID-19 pandemic.

Starting from our results, future studies should focus particularly on undergraduates and gamers, in order to analyze which variables prevent PG, also informing clinicians about intervention programs aimed at preventing or treating PG. Lastly, our findings underscore the role of situational variables in PG, revealing that less stringent government measures predicted a higher prevalence of PG. Future studies should consider situational and lifestyle variables when studying this topic.

## Figures and Tables

**Figure 1 healthcare-11-03176-f001:**
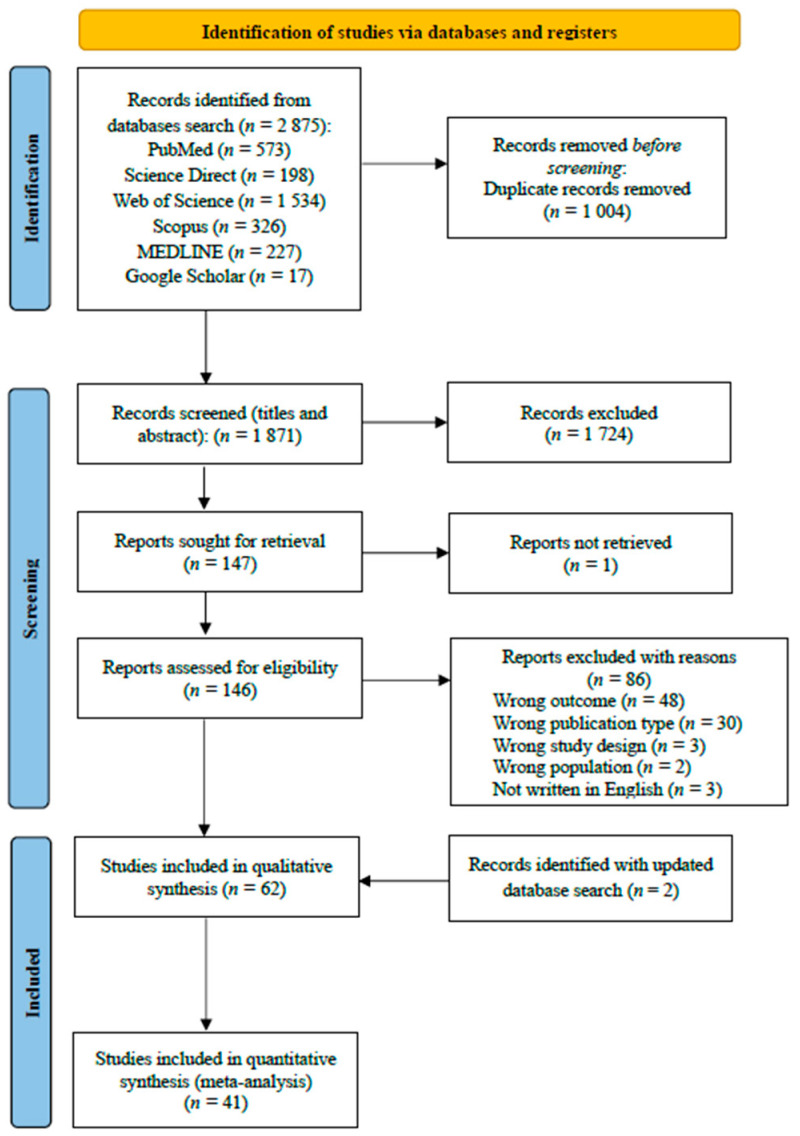
Flow diagram of identification and selection of included studies.

**Figure 2 healthcare-11-03176-f002:**
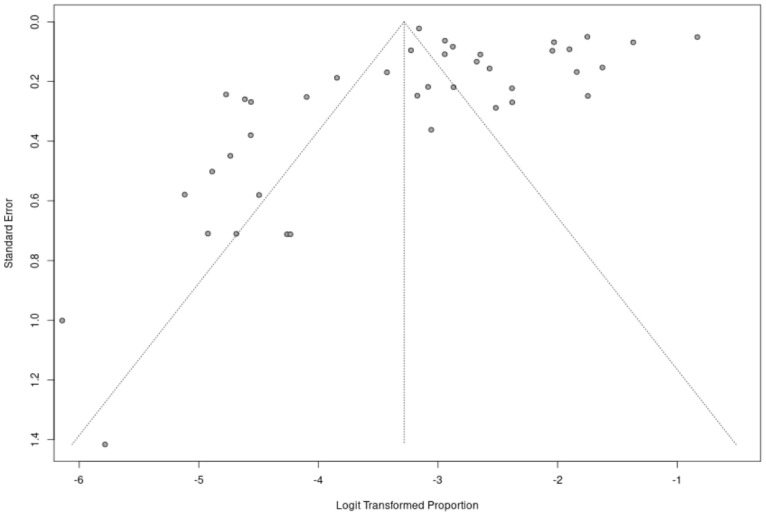
Funnel plot illustrating the distribution of prevalence estimates from included studies. Each point represents an individual study, and the spread of points reflects the precision of estimates.

**Table 1 healthcare-11-03176-t001:** Study characteristics.

Author(s)	Country	Sample (N)	Female (%)	Age (M or Range)	Type of Population	Measure	Risk of Bias	Metanalysis
Alqassim et al. (2022) [28]	Saudi Arabia	427	57.80	16.84	Students	VASC	Low	Yes
Al-Sharqi and Hasan (2022) [29]	World	2526	na	7–23	General	Ad-hoc questionnaire	High	No
Balhara et al. (2020) [17]	India	128	60.00	19.60	Undergraduate	IGDS9-SF	Low	Yes
Brailovskaia et al. (2022) [30]	Germany	131	27.90	25.10	General	IGDS9-SF	Low	Yes
Cakiroglu et al. (2021) [31]	Turkey	410	54.58	13.70	Students	IGD-20	Low	Yes
Cena et al. (2022) [32]	Italy	502	67.70	15.90	Students	VGS	Low	Yes
Chang et al. (2022) [33]	China	1305	41.50	15.16	Students	DSM5-IGD-9	Low	Yes
C.-Y. Chen, Chen, O’Brien, et al. (2021) [34]	China	1357	51.00	10.67	Students	IGDS9-SF	Low	Yes
I. H. Chen et al. (2021) [35]	China	504	51.30	11.29	Students	IGDS9-SF	Low	Yes
I.-H. Chen et al. (2021) [36]	China	535	51.00	10.32	Students	IGDS9-SF	Low	Yes
C.-Y. Chen, Chen, Pakpour, et al. (2021) [37]	China	2026	50.00	10.71	Students	IGDS9-SF	Low	Yes
I. Chen et al. (2022) [38]	China	980	83.00	34.76	General	IGDS9-SF	Low	Yes
C.-Y. Chen et al. (2022) [39]	China	575	50.00	10.83	Students	IGDS9-SF	Low	Yes
I.-H. Chen et al. (2022) [40]	China	272	0	10.92	Students	IGDS9-SF	Low	Yes
Ciccarelli et al. (2022) [41]	Italy	466	45.50	22.24	General	IGDS9-SF	Low	Yes
Claesdotter-Knutsson et al. (2022) [42]	Sweden	932	48.50	16–39	General	GASA-7	Low	Yes
Cuong et al. (2021) [43]	Vietnam	2084	50.20	14.50	Students	IGD-20	Low	Yes
De Pasquale et al. (2021) [44]	Italy	162	51.90	9.40	Students	VASC	Low	Yes
Donati et al. (2021) [45]	Italy	554	79.00	45.52	General	VGS	Low	No
Efrati and Spada (2022) [46]	Israel	2074	60.22	16.14	Students	SSBA	Low	Yes
Elhai et al. (2021) [47]	North America	812	50.10	44.45	General	IGD-4	Low	No
Elsayed (2021) [48]	UAE	289	35.30	6–17	Students	Ad-hoc questionnaire	High	No
Fernandes et al. (2020) [49]	World	188	65.76	21.59	Students	GAS-7	Low	No
Giardina et al. (2021) [7]	Italy	664	8.70	23.59	Gamers	IGD-10	Low	No
Gómez Galán et al. (2021) [50]	Spain	310	69.90	23.70	Undergraduate	GAS-7	Low	Yes
Hall et al. (2021) [51]	World	1144	43.62	31.40	General	DSM5-IGD-9	High	Yes
Higuchi et al. (2020) [52]	Japan	80	2.50	18.90	Gamers	Interviews	High	No
Ismail et al. (2021) [53]	Malaysia	237	69.60	21.50	Students	IGDS9-SF	Low	No
Kim et al. (2021) [54]	South Korea	2984	48.10	13.60	Students	MGUS	Low	Yes
Kim and Lee (2021) [55]	South Korea	2906	48.50	13.62	Students	MGUS	Low	No
Koós et al. (2022) [56]	Hungary	1043	49.50	41.96	General	DSM5-IGD-10	Low	Yes
Maraz et al. (2021) [57]	USA	1430	39.00	36.60	General	Ad-hoc questionnaire	Low	No
Müller et al. (2022) [58]	Europe	174	82.00	20.57	General	DSM5-IGD-10	Low	Yes
Naaj and Nachouki (2021) [59]	UAE	418	55.26	na	Students	Ad-hoc questionnaire	High	No
Nugraha et al. (2021a) [60]	Indonesia	136	36.76	16.02	Students	GAS-7	Low	Yes
Nugraha et al. (2021b) [61]	Indonesia	1046	61.76	15.94	Students	GAS-7	Low	Yes
Oka et al. (2021) [62]	Japan	51,246	50.10	46.60	General	DSM5-IGD-9	Low	Yes
Oliveira et al. (2022) [63]	Brazil	329	90.00	39.60	General	GAS-21	Low	No
Paschke et al. (2021) [64]	Germany	1221	46.11	10–17	Students	GADIS-A-10	Low	No
Rodda et al. (2022) [65]	New Zealand	93	63.00	44.00	General	Ad-hoc questionnaire	High	No
Rozgonjuk et al. (2022) [66]	World	299	16.50	24.37	Gamers	IGDS9-SF	High	Yes
Sallie et al. (2021) [67]	United Kingdom	1344	24.18	28.93	General	IGDS9-SF	Low	Yes
Saritepeci et al. (2022) [68]	Turkey	588	69.60	21.35	Students	MSOGA	Low	No
Savolainen et al. (2022) [69]	Finland	1530	49.67	46.67	General	DSM5-IGD-10	Low	Yes
She et al. (2021) [70]	Hong Kong	3136	51.90	13.60	Students	DSM5-IGD-9	Low	Yes
She et al. (2022) [71]	Hong Kong	3136	51.90	13.60	Students	Ad-hoc questionnaire	Low	No
Shrestha et al. (2020) [72]	Nepal	260	52.30	20.85	Undergraduate	IGDS9-SF	Low	Yes
Singh et al. (2022) [73]	India	1027	58.42	13–60	General	DSM5-IGD-9	Low	No
Son et al. (2021) [74]	South Korea	77	31.20	21.20	Students	IGD-27	Low	No
Teng et al. (2021) [18]	China	1778	49.30	12.20	Students	IGDS9-SF	Low	Yes
Ting and Essau (2021) [75]	Malaysia	178	82.00	22.56	Undergraduate	GAS-7	Low	Yes
Tzang et al. (2022) [76]	Taiwan	102	31.37	7–18	Students	CIAS	Low	No
Volpe et al. (2022) [77]	Italy	1385	62.50	32.50	General	IGDS9-SF	Low	Yes
Wang et al. (2022) [78]	China	324	49.70	13.07	Students	DSM5-IGD-10	Low	No
Werling et al. (2022) [79]	Switzerland	454	na	na	General	Ad-hoc questionnaire	High	No
Wu et al. (2022) [80]	China	5268	47.40	27.00	Gamers	IGDS9-SF	Low	Yes
Xiang et al. (2022) [81]	China	1023	50.64	13.60	Students	IGDQ-11	Low	No
Yang et al. (2021) [82]	Hong Kong	177	52.50	18+	Gamers	DSM5-IGD-9	High	Yes
Yao et al. (2021) [83]	Malaysia	163	50.30	22.43	Students	IGD-10	Low	No
Zaman et al. (2022) [84]	Pakistan	618	32.50	24.53	General	GAS-7	Low	Yes
Zarco-Alpuente et al. (2021) [85]	Spain	1275	66.10	26.23	General	Ad-hoc questionnaire	Low	No
Zhu et al. (2021) [86]	Hong Kong	2848	52.46	12.60	Students	GAS-7	Low	Yes

Note: na: Data not available. Measures: VASC, Video games Addiction Scale for Children; IGD, Internet Gaming Disorder; VGS, Video Gaming Scale; DSM-IGD, Internet Gaming Disorder based on DSM5; GAS, Gaming Addiction Scale; SSBA, Screener for Substance and Behavioral Addictions; MGUS, Maladaptive Game Use Scale; GADIS, Gaming Disorder Scale for Adolescents.

**Table 2 healthcare-11-03176-t002:** Meta-analysis results.

	Sub-Groups Analysis					Meta-Regressions		
	k	Proportion	95%CI	*I* ^2^	*Q (df*)	Estimate	*SE*	*t* (*df*)
*Study location*					6.16 (3)			
Middle East	3	0.092	0.007, 0.590	98.8%				
Asia	23	0.040	0.024, 0.065	98.7%				
Europe	13	0.021	0.010, 0.046	93.7%				
World	2	0.066	0.000, 0.989	97.7%				
*Categories of participants*					22.23 (3) ***			
Students	20	0.035	0.018, 0.066	98.6%				
Undergraduate	4	0.102	0.046, 0.214	83.6%				
General	14	0.023	0.014, 0.040	91.2%				
Gamers	3	0.082	0.026, 0.227	95.0%				
*Rating scale*					16.78 (3) ***			
GAS	7	0.085	0.056, 0.127	94.4%				
IGDS	19	0.024	0.013, 0.043	95.9%				
DSM5	10	0.038	0.016, 0.085	99.3%				
Others	5	0.041	0.006, 0.242	99.3%				
*Risk of bias*					3.05 (1)			
Low	38	0.034	0.022, 0.051	98.8%				
High	3	0.071	0.014, 0.293	95.6%				
Average age						−0.013	0.019	−0.690 (37)
Gender distribution						1.506	1.138	1.323 (39)
Days passed since the beginning of COVID-19						0.002	0.001	1.442 (37)
GSI						−4.481	1.527	−2.935 (37) **
New cases per million people						0.000	0.002	0.162 (37)
New deaths per million people						0.000	0.000	0.502 (37)

Note: ** *p* < 0.01; *** *p* < 0.001.

## Data Availability

Data are reported in both Appendix A and OSF platform (https://osf.io/uvwy9.).

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
