# Peer review of "Problematic Gaming during COVID-19 Pandemic: A Systematic Review, Meta-Analysis, and Meta-Regression"

_healthcare, 2023, doi:10.3390/healthcare11243176_

Round 1

Reviewer 1 Report

Comments and Suggestions for Authors

In the current study, the authors set out to examine problematic gambling behavior during the COVID-19 Pandemic. Research examining factors involved in behavioral addiction and treatment is important, however the manuscript could be improved by addressing the following concerns.

Introduction:

-The manuscript would benefit from better subheadings and generally improved organization of the introduction. It may be helpful to discuss problematic gaming generally and what is known, before discussing impact of the pandemic.

Materials and Methods:

-The manuscript would benefit from an update to the literature search as the paper is now being reviewed nearly a year after the search was conducted.

-The manuscript would benefit from explanation as to why studies were included in review, but excluded from meta-analysis.

-The manuscript would benefit from discussion of what was extracted from each manuscript eligible for meta-analysis.

Results:

-Watch formatting of tables.

-It is confusing as to why students and undergraduates were not included in the same subgroup.

Author Response

We thank the reviewer for their valuable suggestion. Please, find our responses to reviewer's comments point by point in red. 

In the current study, the authors set out to examine problematic gambling behavior during the COVID-19 Pandemic. Research examining factors involved in behavioral addiction and treatment is important, however the manuscript could be improved by addressing the following concerns.

Introduction:  -The manuscript would benefit from better subheadings and generally improved organization of the introduction. It may be helpful to discuss problematic gaming generally and what is known, before discussing impact of the pandemic.

Authors: We thank the reviewer for their comment, we changed the structure of the Introduction part by adding sub-heading in order to improve its clarity.

Materials and Methods: -The manuscript would benefit from an update to the literature search as the paper is now being reviewed nearly a year after the search was conducted.

Authors: Thank you for highlighting this aspect. The research was conducted approximately a year ago, and we cannot rule out the possibility of new contributions being eligible. However, conducting a new search on the databases would entail carrying out a fresh meta-analysis, and more importantly, the global impact of COVID-19 is no longer of the same magnitude as in 2020 and 2021. For these reasons, we believe that conducting the research again in the databases runs the risk of further extending timelines, diminishing the novelty and impact of our meta-analysis, with the potential of adding little to what has already emerged."

-The manuscript would benefit from explanation as to why studies were included in review, but excluded from meta-analysis.

Authors: Thank you for your suggestion. We added one sentence specifying that we included in the meta-analysis only quantitative studies providing sufficient data to compute analysis, whereas we included in the systematic review eligible studies that not provided sufficient data.

-The manuscript would benefit from discussion of what was extracted from each manuscript eligible for meta-analysis.

Authors: Thank you for your suggestion, we included a further supplemental table including data extracted from each study along with the sample size, in order to improve the reproducibility of the present metanalysis.

Results: -Watch formatting of tables.

Authors: We checked the journal’s guidelines on formatting tables, and tables included in the current form are in lines with recommendations.

-It is confusing as to why students and undergraduates were not included in the same subgroup.

Authors: Thank you for your feedback. Our decision to differentiate between these groups was based on the distinct nature of their experiences and perspectives given the different age. Indeed, among the students, we considered individuals attending elementary, middle, and high schools, while among the undergraduates, we included individuals enrolled in university who have not yet graduated. We specified this aspect in the paragraph 2.5 Data collection and statistical analysis.

Reviewer 2 Report

Comments and Suggestions for Authors

The present manuscript is a meta-analysis on the prevalence of problematic gaming during the COVID-19 pandemic, as well as some potential moderating variables of that prevalence. Despite the existence of several systematic reviews and similar meta-analyses, the authors adequately justify the need for the study they present.

In general, the introduction is correctly written, well-structured, and provides a precise summary of both problematic gaming and how it may have been affected by the pandemic.

Regarding the method, it would be interesting to know what percentage of authors responded to the data request. Furthermore, it would be advisable to provide a more detailed explanation of the data extraction process. For instance, was it conducted by a single researcher? In the event that it was performed by more than one researcher, what was the agreement percentage, and how were discrepancies resolved?

I think it would be prudent to conduct a sensitivity analysis to examine if any study is exerting excessive influence on the estimated prevalence.

It is advisable to exercise caution when interpreting the results of subgroup analyses, as the number of studies in some of the groups is very low (n=2 or n=3), which could result in low statistical power.

In the limitations section, and despite the fact that the majority of scientific literature is published in English, I believe it would be advisable to add that relevant studies published in different languages may have been excluded.

Author Response

We thank the reviewer for their valuable suggestions. Please, find a point-by-point response to the comments in red.

The present manuscript is a meta-analysis on the prevalence of problematic gaming during the COVID-19 pandemic, as well as some potential moderating variables of that prevalence. Despite the existence of several systematic reviews and similar meta-analyses, the authors adequately justify the need for the study they present.

In general, the introduction is correctly written, well-structured, and provides a precise summary of both problematic gaming and how it may have been affected by the pandemic.

Authors: We thank the reviewer for their positive comment.

Regarding the method, it would be interesting to know what percentage of authors responded to the data request.

Authors: Thank you for noting this aspect. Authors contacted were included in the supplementary materials, but unfortunately we failed to upload such files in the system. We trust that everything is functioning correctly with the current submission. By the way, we contacted a total of 17 authors, and 7 authors did not responded providing data.

Furthermore, it would be advisable to provide a more detailed explanation of the data extraction process. For instance, was it conducted by a single researcher? In the event that it was performed by more than one researcher, what was the agreement percentage, and how were discrepancies resolved?

Authors: In line with PRISMA guidelines, one author performed the preliminary scrutiny on titles and abstracts aiming at excluding irrelevant references. Then, two different authors independently assessed the full texts of the remaining studies based on our inclusion and exclusion criteria. Lastly, we compared results and disagreements were resolved by mutual consensus. Please, see paragraph 2.3 Selection process. Once the studies were selected, two authors extracted the data necessary for calculating prevalence, conducting subgroup analyses and meta-regressions.

I think it would be prudent to conduct a sensitivity analysis to examine if any study is exerting excessive influence on the estimated prevalence.

Authors: We thank the reviewer for the suggestion. Unfortunately, as this is a prevalence meta-analysis and not focused on effect sizes, we are unable to conduct the sensitivity analysis. However, we have performed an analysis excluding potential outliers, and these are the results (random effects model, k = 21): prevalence = 0.0325, 95%CI [0.0239, 0.0440], I^2 = 80.6%, Q = 103.00, p < .001. Therefore, outlier analysis confirmed the order of magnitude and its significance.

It is advisable to exercise caution when interpreting the results of subgroup analyses, as the number of studies in some of the groups is very low (n=2 or n=3), which could result in low statistical power.

Authors: Thank you for your feedback, we appreciate your concern regarding the subgroup analyses, particularly in groups with a low number of studies (n=2 or n=3). We acknowledge the potential limitations associated with small frequencies and the impact on statistical power. We included one sentence in the limitation section, please, see the last paragraph of the Discussion section.

In the limitations section, and despite the fact that the majority of scientific literature is published in English, I believe it would be advisable to add that relevant studies published in different languages may have been excluded.

Authors: We added one sentence among limitations specifying this aspect, thank you for suggesting it.

Reviewer 3 Report

Comments and Suggestions for Authors

The manuscript estimated the prevalence of PG during the COVID

19 and examined differences between subpopulations. Evaluation of 38 studies revealed that the overall prevalence of PG during the COVID 19 pandemic was 3.6%.

The results indicate that higher PG scores were found in subpopulations of college students and gamblers, as well as in studies that used the Gambling Addiction Scale.

The authors conclude that regression meta-analyses suggest that stricter government measures, as identified by the Government Stringency Index, may have contributed to a lower prevalence of PG behaviors.

This study sheds light on a possible explanation for this finding is that restraint measures had a protective function with respect to emotional distress and thus toward GP; alternatively that current measures for GP become less accurate if individuals' functioning is already impaired for other reasons, such as COVID 19 restrictions.

Authors should explicitly include, in the introduction, the question to be answered.

It is also appropriate for the authors to include in the introduction some reference to the Meta-analysis of reliability generalization of the Internet gaming disorder scale.

It is important that the authors refer to the psychosocial risk factors of technological addictions: the influence of emotional (dys)regulation, personality traits and the fear of missing out on something in Internet addiction.

Along these lines, it is appropriate to address the role of emotional (dys)regulation in Internet addiction.

The method is well described.

I would strongly appreciate it if the authors would address, in the discussion, the cognitive, social and health-related factors that explain gambling addiction. Gambling motives, cognitive distortions, and irresponsible gambling, an explanatory model of gambling addiction, should also be addressed.

It would be desirable to include in the introduction some citation, and later some reference, to a systematic review addressing problematic Internet use and resilience.

Conclusions are strong with the evidence and arguments presented.

Citations and references are appropriate and up to date, but authors should review the journal's guidelines, so that they are as requested in Healthcare.

Tables and figures are very clear.

Data are reproducible.

The authors contribute knowledge to science with this manuscript, being relevant and of interest.

Author Response

We thank the reviewer for their valuable suggestions. Please, find a point-by-point response to the comments in red.

The manuscript estimated the prevalence of PG during the COVID19 and examined differences between subpopulations. Evaluation of 38 studies revealed that the overall prevalence of PG during the COVID 19 pandemic was 3.6%.

The results indicate that higher PG scores were found in subpopulations of college students and gamblers, as well as in studies that used the Gambling Addiction Scale.

The authors conclude that regression meta-analyses suggest that stricter government measures, as identified by the Government Stringency Index, may have contributed to a lower prevalence of PG behaviors.

This study sheds light on a possible explanation for this finding is that restraint measures had a protective function with respect to emotional distress and thus toward GP; alternatively that current measures for GP become less accurate if individuals' functioning is already impaired for other reasons, such as COVID 19 restrictions.

Authors should explicitly include, in the introduction, the question to be answered.

Authors: We thank the reviewer for their comment. We added one sentence clearly specifying the aim of our study, see paragraph 1.2 Objectives of the present systematic review and meta-analysis.

It is also appropriate for the authors to include in the introduction some reference to the Meta-analysis of reliability generalization of the Internet gaming disorder scale.

Authors: Thank you for your suggestion, we added the reference to the meta-analysis you mentioned in the Discussion part, when discussing results on scales used to assess PG.

It is important that the authors refer to the psychosocial risk factors of technological addictions: the influence of emotional (dys)regulation, personality traits and the fear of missing out on something in Internet addiction.

Along these lines, it is appropriate to address the role of emotional (dys)regulation in Internet addiction.

Authors: We appreciate the suggestion, and in light of it, we have specifically focused on exploring psychosocial risk factors associated with problematic gaming.

The method is well described.

I would strongly appreciate it if the authors would address, in the discussion, the cognitive, social and health-related factors that explain gambling addiction. Gambling motives, cognitive distortions, and irresponsible gambling, an explanatory model of gambling addiction, should also be addressed.

It would be desirable to include in the introduction some citation, and later some reference, to a systematic review addressing problematic Internet use and resilience.

Authors: We added the reference to the systematic review you mentioned in the Discussion, linking our results to some of the topics emerged in the review.

Conclusions are strong with the evidence and arguments presented.

Citations and references are appropriate and up to date, but authors should review the journal's guidelines, so that they are as requested in Healthcare.

Tables and figures are very clear. Data are reproducible.  

The authors contribute knowledge to science with this manuscript, being relevant and of interest.

Authors: We thank the reviewer for their positive comments, we changed the reference style following the journal’s guidelines.

Reviewer 4 Report

Comments and Suggestions for Authors

1. Long paragrpaphs are unwanted. Please restructure the narration into smaller paragraphs.

2. Titles of measures (in tables) should be deciphered in the notes for these tables. Table 1, Werling et al. (2022): what measure did these authors use here? "Questionaire"?

3. "Contrary to what we expected, the relationship between PG and GSI was negative, such that the fewer the measures adopted by the countries, the higher was PG prevalence".

What did you expected? Please could you present your hypotheses of the study?

4. Supplementary Materials and Appendices were not presented. Please make sure that all materials are here before second review. All these materials should be peer-reviewed too.

Author Response

We thank the reviewer for their valuable suggestions. Please, find a point-by-point response to the comments in red.

  1. Long paragrpaphs are unwanted. Please restructure the narration into smaller paragraphs.

Authors: We changed the Introduction structure dividing it into smaller paragraphs, thank you for suggesting it.

  1. Titles of measures (in tables) should be deciphered in the notes for these tables. Table 1, Werling et al. (2022): what measure did these authors use here? "Questionaire"?

Authors: We added one note in Table 1 specifying the titles of measures, and corrected the measure used by Werling et al 2022, thank you for suggesting

  1. "Contrary to what we expected, the relationship between PG and GSI was negative, such that the fewer the measures adopted by the countries, the higher was PG prevalence".

What did you expected? Please could you present your hypotheses of the study?

Authors: We apologize for the lack of clarity in this sentence. In this study, we did not have a specific hypothesis regarding this analysis. What we meant to convey is that it was reasonable to expect a positive relationship between PG and GSI, i.e., the more stringent the government-imposed rules, and the more people had to stay isolated at home, the higher the likelihood of developing PG. However, our results demonstrated the opposite relationship. For clarity, we have removed the phrase 'contrary to what we expected.'

  1. Supplementary Materials and Appendices were not presented. Please make sure that all materials are here before second review. All these materials should be peer-reviewed too.

Authors: We are sorry for this oversight, unfortunately we failed to upload such files in the system. We trust that everything is functioning correctly with the current submission.

Round 2

Reviewer 1 Report

Comments and Suggestions for Authors

Authors sufficiently addressed concerns previously mentioned.

Reviewer 4 Report

Comments and Suggestions for Authors

Thanks for improvments. The paper is improved, and ready for publication.